# Regulatory Effects of the *Kiss1* Gene in the Testis on Puberty and Reproduction in Hezuo and Landrance Boars

**DOI:** 10.3390/ijms242316700

**Published:** 2023-11-24

**Authors:** Haixia Shi, Zunqiang Yan, Hong Du, Yuran Tang, Kelin Song, Qiaoli Yang, Xiaoyu Huang, Pengfei Wang, Xiaoli Gao, Jiaojiao Yang, Shuangbao Gun

**Affiliations:** 1College of Animal Science and Technology, Gansu Agricultural University, Lanzhou 730070, China; 2Gansu Research Center for Swine Production Engineering and Technology, Lanzhou 730070, China

**Keywords:** Hezuo boars, Landrace boars, *Kiss1*, testis, puberty, spermatogenesis

## Abstract

Kisspeptin, a neuropeptide encoded by the *Kiss1* gene, combines with its receptor Kiss1R to regulate the onset of puberty and male fertility by the hypothalamic–pituitary–gonadal axis. However, little is known regarding the expression signatures and molecular functions of *Kiss1* in the testis. H&E staining revealed that well-arranged spermatogonia, spermatocytes, round and elongated spermatids, and spermatozoa, were observed in 4-, 6-, and 8-month-old testes compared to 1- and 3-month-old testes of Hezuo pigs; however, these were not observed in Landrance until 6 months. The diameter, perimeter, and cross-sectional area of seminiferous tubules and the perimeter and area of the tubular lumen increased gradually with age in both pigs. Still, Hezuo pigs grew faster than Landrance. The cloning results suggested that the Hezuo pigs’ *Kiss1* CDS region is 417 bp in length, encodes 138 amino acids, and is highly conserved in the kisspeptin-10 region. qRT-PCR and Western blot indicated that the expression trends of *Kiss1* mRNA and protein were essentially identical, with higher expression levels at post-pubertal stages. Immunohistochemistry demonstrated that the Kiss1 protein was mainly located in Leydig cells and post-pubertal spermatogenic cells, ranging from round spermatids to spermatozoa. These studies suggest that Kiss1 is an essential regulator in the onset of puberty and spermatogenesis of boars.

## 1. Introduction

Puberty is an intricate physiological process in which an organism attains sexual maturity, characterized by the acquisition of secondary sexual characteristics, the maturation of the gonads, as well as the attainment of reproductive capacity [1,2,3]. From the anatomical perspective of the gonads (e.g., testes), the seminiferous tubules in the testes are nearly parenchymal with no obvious lumen, and the development of spermatogenic cells is at a standstill until puberty when seminiferous tubule lumen and all levels of spermatogenic cells, from spermatogonia to spermatozoa, can be discernable [4,5]. In addition, the diameter and area of the seminiferous tubules, and the area and perimeter of the tubule lumens gradually increased with testicular development [6,7]. Puberty emerges as a result of complex neuroendocrine mechanisms that involve the maturation and activation of the hypothalamic–pituitary–gonadal (HPG) axis [8,9]. Formed in hypothalamic neurons, gonadotropin-releasing hormone (GnRH), a decapeptide central to the initiation of the reproductive hormone cascade, is released in a pulsatile manner into the hypophyseal portal circulation to stimulate the biosynthesis and secretion of luteinizing hormone (LH) and follicle-stimulating hormone (FSH) from the pituitary, which subsequently acts on the gonads to promote gametogenesis and the production of sex steroids [1,10]. In turn, the release of GnRH itself is regulated by excitatory and inhibitory signals in the form of neurohormones and neurotransmitters acting at the level of the hypothalamus [11].

Kisspeptin, encoded by the *Kiss1* gene, was initially recognized as a metastasis suppressor [12,13]. Current evidence indicates that kisspeptin and its G protein-coupled receptor 54 (GPR54/Kiss1R), localized in the hypothalamus, play pivotal roles in the onset of puberty and adult reproduction in mammals by modulating the HPG axis [14,15,16,17]. Disruption of kisspeptin signaling results in diverse reproductive disorders. For illustration, knockout rats with *Kiss1* lack pulsatile and proliferative patterns of gonadotropin and display puberty failure [18]; in contrast, mutations leading to hyperactive *Kiss1* result in central precocious puberty in humans [19,20]. In addition, central administration of kisspeptins in immature female rats could induce precocious activation of the HPG axis [21]. In conclusion, these results illustrate that kisspeptin in the hypothalamus exerts significant effects on the onset of puberty.

There is no doubt about the central role of kisspeptin in stimulating the release of GnRH from the hypothalamus. Much less is known, however, about the role of kisspeptin in gonadal tissues such as the testis. So far, *Kiss1* and its protein have been detected in the testes of mice [22,23], goats [24], chub mackerel [25], rhesus monkeys [26], and humans [27], demonstrating that kisspeptin has an autocrine or paracrine function in the testis. Nevertheless, the expression and distribution of *Kiss1* and its proteins differ in testicular cells of different species. For example, expression of *Kiss1* and its protein was observed in human sperm cells [28], mouse Leydig cells [22,29], and goat Leydig cells and spermatids [24,30]. Furthermore, kisspeptin was immunolocalized in mouse [14,22,23] and goat [24,31] Leydig cells, rhesus monkey spermatocytes and spermatids [26], and human sperm cells [28]. The expression mode of *Kiss1* has been described in testis development and spermatogenesis. *Kiss1* expression is higher in pubertal testes than in reproductively active, birth, and pre-pubertal testes [22,24]. Moreover, one report shows that kisspeptin in the testis of scombroid fish may be related to spermatogenesis and gonadal development [32].

Hezuo (HZ) pigs, a famous native pig breed, reach puberty at a relatively younger age than the Landrace (LC) [33] pigs. However, little is understood about the expression profiles and biological functions of Kiss1 in the HZ pig testes. Hence, the current work was undertaken: (i) to observe and compare the histological characteristics of HZ boars and LC boars’ testes; (ii) to gain the coding sequence (CDS) of the *Kiss1* gene in HZ pigs, and to analyze its molecular features; (iii) to explore and compare expression profiles and immunolocalization of Kiss1 during testicular development and spermatogenesis in HZ and LC boars. This study will provide new insights into the mechanism of action of *Kiss1* during the onset of puberty and spermatogenesis in boars.

## 2. Results

### 2.1. Morphological Analysis of Testicular Tissue from HZ and LC Boars

To better understand testicular development, we harvested testicular tissues from HZ and LC boars at 1, 3, 4, 6, and 8 months of age, and subsequently observed the histomorphology of the testes by H&E staining, as shown in Figure 1A. The results demonstrate that testicular tissues mainly consist of a variety of germ cells and Sertoli cells within the seminiferous tubules and Leydig cells between the seminiferous tubules. However, the timing of the appearance of the various germ cells varies considerably between HZ and LC boars. Only spermatogonia was detected in the seminiferous tubules of the HZ and LC boars in the 1 M (month) group. In addition to spermatogonia, a small number of primary spermatocytes, secondary spermatocytes, and elongated spermatids were found in the seminiferous tubules of 3 M HZ boars, but not in 3 M LC boars. A significant increase in the spermatogenic cell layer was observed and all levels of spermatogenic cells (spermatogonia, primary spermatocytes, secondary spermatocytes, round spermatids, elongated spermatids, and spermatozoa) were sequentially arranged from the basal compartment to lumen of seminiferous tubules in 4 M HZ boars, but no late-developing spermatogenic cells (round spermatids, elongated spermatids, and spermatozoa) were seen in 4 M LC boars until 6 M. Finally, more spermatogenic cells were identified in the seminiferous tubules of HZ and LC boars at the 8 M group.

A detailed analysis was conducted to determine the morphological parameters in the cross sections of 45 seminiferous tubules from both breeds at each age group, as presented in Figure 1B. In HZ and LC boars, morphological parameters, such as the diameter, perimeter, and cross-sectional area of the seminiferous tubules, and the perimeter and area of the tubular lumen, progressively enlarged with age. Comparison of the same developmental period in both pig breeds revealed extremely significant differences between groups (*p* < 0.01), and it is noteworthy that the values of the parameters (the perimeter and area of the tubular lumen) in HZ boars were greater than those in LC boars in the 1 M, 3 M, 4 M, and 6 M groups, and stabilized in the 4 M, 6 M, and 8 M groups. In addition, the ratio of the lumen area to the cross-sectional area of seminiferous tubules was highest in the 4 M group of HZ boars compared to the other groups (*p* < 0.01).

### 2.2. CDS Sequence Characteristics of HZ Boars Kiss1

The RT-PCR amplification product of the *Kiss1* gene from HZ boars was detected by 1.5% agarose gel electrophoresis, obtaining a specific band of approximately 433 bp (Figure 2A), which was consistent with the expected target fragment. The cloned *Kiss1* cDNA sequence included 417 bp, encoding 138 amino acids (Figure 2B). Compared to the porcine reference sequence (Accession No. NM_001134964.1), there is a base mutation (C → T) in the CDS sequence of the cloned HZ porcine *Kiss1* gene at nucleotide position 121, resulting in a proline mutation to a serine (Figure 2C). Amino acid sequence comparison showed that HZ boars Kiss1 amino acid sequence displayed 99.28% sequence similarity with pig, 51.34% with cattle, 69.57% with goat, 67.39% with sheep, and 52.17% with rabbit *Kiss1* (Figure 2D). Interestingly, kisspeptin is highly conserved in the kisspeptin-10 region across species.

### 2.3. Physicochemical Properties and Structure Analysis of HZ Boars Kiss1 Protein

The molecular formula, theoretical isoelectric point (pI), and molecular weight of the HZ boars Kiss1 protein were C_667_H_1045_N_203_O_192_S_6_, 10.35, and 15,172.24 Da, respectively. The number of amino acids encoded by the *Kiss1* gene was 138, of which proline was the largest, accounting for 13.8% of the total (Figure 3A). The predicted secondary structure of the Kiss1 protein revealed it to be a protein with a mixed secondary structure consisting of 72.46% random coil, 19.57% α-helix, 6.52% extended strand, and 1.45% β-turn (Figure 3B). The predicted tertiary structure of the Kiss1 protein is predominantly random coil, which is generally consistent with the predicted results of the secondary structure (Figure 3C). The Kiss1 protein contains a signal peptide located at positions 1 to 17 between amino acid residues (Figure 3D) and is devoid of transmembrane structure (Figure 3E). The protein encoded by HZ boars *Kiss1* may interact with 10 proteins, including its receptor and G protein-coupled receptor 54 (Kiss1R) (Figure 3F). The Kiss1 protein has a kisspeptin-conserved domain, which belongs to the kisspeptin superfamily (Figure 3G).

### 2.4. Expression Patterns of Kiss1 at the Transcript and Protein Levels in Developmental Testes of HZ and LC Boars

qRT-PCR analysis revealed that *Kiss1* mRNA was expressed in all five stages of testicular tissues in both HZ and LC boars, with its expression gradually increasing with age. However, the expression characteristics were not identical in both breeds of boars (Figure 4A). Specifically, in testicular tissues of HZ boars, *Kiss1* mRNA expression was significantly increased in the 4 M, 6 M, and 8 M groups compared to those in the 1 M and 3 M groups (*p* < 0.01), but in LC boars, it was significantly increased in the 6 M and 8 M groups compared to those in the 1M, 3 M, and 4 M groups (*p* < 0.01) and remained at a stable level (*p* > 0.05). Furthermore, comparisons of two pig breeds at the same developmental stage revealed that *Kiss1* mRNA has extremely significant differences in the 6 M and 8 M groups (*p* < 0.01). Kiss1 protein was detectable in HZ and LC boar testes throughout the developing stages (Figure 4B,C). The trend of Kiss1 protein expression was generally consistent with that of its mRNA (Figure 4D).

### 2.5. Immunolocalization of Kiss1 Protein in Developmental Testes of HZ and LC Boars

To evaluate the positive signals for the Kiss1 protein, we performed immunohistochemistry with Kiss1 primary antibody on testicular tissue sections derived from HZ and LC boars at different age groups, representative images of which are shown in Figure 5. The results revealed that positive signals of the Kiss1 protein were observed in testes of all ages, but its distribution patterns vary with developmental stages and species. Specifically, in the HZ boar testis, Kiss1 positive signals were detected in the epithelia of seminiferous tubules and Leydig cells in the 1 M group, with relatively low expression (Figure 5(A1)). The positive staining pattern of Kiss1 protein was essentially similar in the 3 M, 4 M, 6 M, and 8 M groups, with strong positive signals in late-developing spermatogenic cells (round spermatocytes, elongated spermatocytes, and spermatids) as well as moderate signals in Leydig cells (Figure 5(A2)–(A5)). In LC boar testes, weak positive signals of the Kiss1 protein were detected in the epithelia of seminiferous tubules and Leydig cells in the 1 M and 3 M groups (Figure 5(C1),(C2)), while intense signals were observed in round spermatocytes, elongated spermatocytes, and spermatozoa in the 4 M, 6 M, and 8 M groups, and moderate signals in Leydig cells (Figure 5(C3)–(C5)).

## 3. Discussion

Although several reproductive traits of pigs have been preserved during evolution, breed variations still exist in the rate of testicular development and the timing of puberty onset. In the present research, we analyzed and compared the morphological characteristics and Kiss1 expression and localization during testicular development in HZ and LC boars, aiming to further understand the differences in testicular development between the two breeds. Previous studies have suggested that boars usually reach puberty between 4 and 6 months [34,35], whereas Chinese native boars enter puberty earlier. Various types of spermatogenic cells and the lumen of seminiferous tubules can be observed in the testicular tissues after the boar reaches puberty [5]. Here, spermatocytes and the lumen of seminiferous tubules were identified in HZ boars’ testes since 3 M, while not detected in LC boars until 4 M. Advanced spermatogenic cells were observed in HZ boars’ testes in the 4 M group, while not found in LC boars until 6 M. These results indicated that sperm production and onset of puberty were earlier in HZ boars compared to LC boars. In addition, the measurement of the morphological parameters of testicular seminiferous tubule suggested that the diameter, perimeter, and cross-sectional area of the seminiferous tubules, and the perimeter and area of the tubular lumen of HZ and LC boars increased with the age of the animals. The values of morphological parameters (the perimeter and area of the tubular lumen) in HZ boars were greater than those in LC boars in the 1 M, 3 M, 4 M, and 6 M groups, suggesting that seminiferous tubular lumen appears earlier in HZ boars than in LC boars. In conclusion, histological analyses revealed that Chinese HZ boars have an earlier onset of puberty and earlier testicular development compared to LC boars [33], which is in agreement with other indigenous porcine breeds [36,37].

Kisspeptin is a neuropeptide produced by the *Kiss1* gene and mature kisspeptins are cleaved into a variety of endogenous fragments, such as kisspeptin-54, -16, -14, -13, and -10 [27,38]. It serves as an essential regulator of puberty onset, sexual maturation, and reproductive functions [39,40,41]. Multiple studies have confirmed that kisspeptin, positioned in the hypothalamus, stimulates testosterone secretion, testis maturation, and spermatogenesis via the HPG axis [42,43,44]. Additionally, a growing number of studies have demonstrated that kisspeptin is also expressed in peripheral tissues such as the testis. However, little is known regarding the differences in the expression and localization of kisspeptin in the testicular tissue of HZ and LC boars. In the present study, molecular cloning indicated that the full-length CDS sequence of the HZ boar *Kiss1* gene was 417 bp and encoded a 138 amino acid precursor protein, which is consistent with prior research on Chinese alligator sinensis [45]. Compared to the nucleotide sequence of the pig reference sequence in the NCBI database, the HZ boar *Kiss1* gene has a base mutation in the sequence of the CDS region, and whether it is associated with precocious puberty in HZ pigs requires further study. Comparison of the deduced amino acid sequence of the HZ boar with other animals revealed that the amino acid sequences were highly conserved only in the kisspeptin-10 region (Kp-10), a feature that has also been noted in previous studies [25,45,46,47]. The kp-10 region of *Kiss1* is reported to encode a core peptide that is the minimal necessary sequence for binding and activating the kisspeptin receptor [27,47,48]. These results suggest that kisspeptin is highly conserved and plays an important role. The physicochemical property analysis revealed that HZ boar *Kiss1* encodes a protein with molecular formula C_667_H_1045_N_203_O_192_S_6_, the molecular weight of 15,172.24 Da, and Pi of 10.35, suggesting that it is a basic protein. The prediction of the signal peptide indicated that the Kiss1 protein contains a signal peptide, indicating that the protein is a secreted protein, which is in general agreement with the findings of Cao et al. [49]. Further analysis indicated that the protein encoded by *Kiss1* in HZ boars may interact with 10 proteins, of which Kiss1R was reported to play a role in the initiation of puberty as its cognate ligand [50,51].

Multiple studies have demonstrated that *Kiss1* mRNA is expressed at higher levels in puberty and adulthood than in prepubertal, correlating with the onset of puberty and spermatogenesis [24]. To explore the expression pattern of the *Kiss1* gene in the testes of HZ and LC boars, and the differences between the two, in the present study, we performed qRT-PCR analysis to assess the expression profiles of the *Kiss1* gene in testes of different developmental phases. The results revealed that *Kiss1* mRNA expression in the testes of HZ boars increases with age, and the expression of *Kiss1* mRNA was relatively higher in the 4 M, 6 M, and 8 M groups compared with the 1 M and 3 M groups, indicating that the *Kiss1* gene plays an important regulatory role in the initiation of puberty and testicular development in HZ boars. This is largely consistent with previous reports on the expression pattern of *Kiss1* in the developing testes of Wuzhishan pigs [52] (an indigenous pig breed with precocious puberty characteristics) and mice [14]. Unlike HZ boars, the expression of *Kiss1* mRNA in the 6 M and 8 M groups of LC boars was relatively higher than that in the 1 M, 3 M, and 4 M groups, probably related to the later initiation of puberty in LC boars. Subsequently, we performed Western blot assay to investigate whether the Kiss1 protein has a similar or identical expression pattern with that at the transcript level. As expected, the Kiss1 protein was expressed at all stages of testicular development in HZ and LC boars, and its expression pattern was essentially identical to that of the *Kiss1* transcript, being significantly more abundant in post-pubertal testes. Based on these results, we speculated that Kiss1 may serve an essential function in testicular development in HZ and LC boars, and that differential temporal expression patterns of Kiss1 may be linked to puberty onset and spermatogenesis.

Although Kiss1 is critical for male testicular development and spermatogenesis, its distribution and function in the testis vary with developmental stages and species. For Shiba goats, positive kisspeptin protein was detected in testicular Leydig cells and presented moderate and intense staining at the pre-pubertal and post-pubertal developmental stages, respectively [24]. In adult male rhesus monkeys, Tariq et al. [26] suggested that the kisspeptin protein was immunolocalized in spermatocytes and spermatids. However, Irfan et al. [53] discovered that the kisspeptin protein was only immunolocalized in the interstitial compartment but not detected in the seminiferous epithelium of adult rhesus monkeys. Moreover, positive signals of the Kiss1 protein were mainly detected in Leydig cells [14,22,23], round spermatids [23], and elongated spermatids [54] of mice. In mouse Leydig cells, kisspeptin immunoreactivity exhibits weak to moderate staining from birth to the prepubertal stage, as well as robust staining during puberty and reproductive activity [22]. In addition, the positive signals for Kiss1 protein were observed in human spermatozoa by Pinto et al. [28]. To elucidate the potential functions of Kiss1 during testicular development in HZ and LC boars and the differences in localization between the two pig breeds, we observed the distribution of the Kiss1 protein in the testes using immunohistochemistry. The results indicated that the positive signals for the Kiss1 protein were observed in HZ and LC boars’ testes at all ages, with moderate staining in epithelia of seminiferous tubules and Leydig cells before puberty, and strong staining in elongated spermatids, round spermatids, and spermatozoa after puberty. The immunolocalization pattern of testicular Leydig cells, elongated spermatids, round spermatids, and spermatozoa in boars is generally consistent with previous findings in Shiba goats [24], mice [14,22,54], and humans [28]. In addition, although the pattern of Kiss1 protein localization was similar in HZ and LC boars, there were differences between the different developmental stages. Kiss1 protein positive signals were detected in elongated spermatids and round spermatids in the 3 M group of HZ boars, but not in LC boars until the 4 M group, possibly because the testes of HZ boars develop faster than those of LC boars, so positive signals of Kiss1 in HZ boars, localized in elongated spermatids and round spermatids, appeared earlier than LC boars.

Spermatogenesis is a highly complex cellular differentiation event, particularly spermiogenesis, the final stage of spermatogenesis, which includes the detachment of cytoplasmic organelles, the formation of long flagellar structures, and intense nuclear rearrangement [55,56]. A study discovered that a subcutaneous injection of Kiss1 pentadecapeptide can accelerate spermatogenesis in prepubertal male chub mackerel [57]. Furthermore, it has also been demonstrated that kisspeptin can induce sperm motility changes and hyperactivation during the late stages of spermatogenesis [28]. The positive signals of Kiss1 protein in this experiment were mainly distributed in round spermatids, elongated spermatids, and spermatozoa of post-pubertal boar testes and Leydig cells at various stages of development, which may act as an important role in the onset of puberty and spermatogenesis, especially during spermiogenesis and sperm motility, but the specific regulatory mechanism remains to be further verified.

## 4. Materials and Methods

### 4.1. Animals and Sample Collection

A total of 15 HZ and 15 LC boars, from five developmental stages—1 M (*n* = 3), 3 M (*n* = 3), 4 M (*n* = 3), 6 M (*n* = 3), and 8 M (*n* = 3)—were obtained from Gansu Sunxiang breeding Co., Ltd. (Zhuoni, Gansu, China). After all boars were humanely slaughtered, the paired testicular tissues of each boar were cleaned and collected. As previously described [5,7,58,59], each testis was cut longitudinally into two slices, and approximately 1-cm^3^ parenchymal sample was collected from the intermediate region between the tunica albuginea and the transitional region (roughly the same area for each testis) and then fixed in Bouin’s solution for hematoxylin and eosin (H&E) staining and immunohistochemistry. Three sections (5 µm) were cut from each tissue, for a total of eighteen sections per experimental group. Subsequently, the remaining testicular tissue was kept at −80 °C for the preparation of total RNA and protein.

### 4.2. H&E Staining of Testis Tissues from HZ and LC Boars

Sections from boars’ testicular tissues at each developmental period were stained using the H&E method. The experimental steps were as follows: (1) sections were dewaxed twice with xylene for 5 min each, dehydrated with 100%, 95%, 85%, and 75% ethanol for 2 min each, and subsequently washed in PBS for 5 min; (2) hematoxylin staining was added dropwise for 5 min, and washed in distilled water for 2 min; (3) differentiated with 1% hydrochloric alcohol for 10 s; (4) 0.5% eosin staining was added dropwise for 2 min and washed in distilled water for 2 min; (5) 95% ethanol soaking twice for 1 min each, anhydrous ethanol soaking twice for 1 min each, and xylene soaking twice for 1 min each; (6) sealing with neutral gum for microscopic observation and image acquisition (Sunny Optical Technology Co., Ltd., Ningbo, China). Subsequently, the morphological parameters of 45 randomly selected round or nearly round seminiferous tubules from the 200× magnification of H&E sections were measured using a Sunny EX31 biological microscope (Sunny, Ningbo, China).

### 4.3. RNA Extraction and cDNA Synthesis

Total RNA was extracted using a Trizol reagent kit (Invitrogen, Carlsbad, CA, USA) as per the manufacturer’s instructions. The purity and concentration of RNA were assessed using an ultra-micro spectrophotometer (Thermo Fisher Scientific, Waltham, MA, USA), and its integrity was checked by 1.5% agarose gel electrophoresis. Then, RNA was reverse transcripted into cDNA by reverse transcription kits (Accurate Biotech, Changsha, China), according to the manufacturer’s instructions.

### 4.4. Cloning of HZ Boars Kiss1 Gene

According to the mRNA sequence of the porcine *Kiss1* gene (Accession No. NM_001134964.1) obtained from the National Center for Biotechnology Information (NCBI), the primer pairs were designed using Primer Premier 5.0 software (Premier Company, Toronto, ON, Canada), and the primers were synthesized by Tsingke Biotechnology Co. Ltd. (Xi’an, China) (Table 1). The cDNA derived from testis tissues of HZ boars served as the template to amplify the CDS sequence of the *Kiss1* gene. The total volume of the PCR reaction was 20 μL, including 10 μL of 2 × Easy Taq PCR SuperMix (TransGen Biotech, Beijing, China), 2 μL of cDNA template, 1 μL of forward primer, 1 μL of reverse primer, and 6 μL of RNase-free H2O. The reaction procedures were as follows: 1 cycle of 95 °C for 5 min; 35 cycles of 95 °C for 30 s, 60 °C for 30 s, and 72 °C for 1 min; and 1 cycle of 72 °C for 10 min. The PCR products were separated using 1.5% agarose gel and purified using an agarose gel DNA recovery kit (TIANGEN, Beijing, China). The purified product was then ligated to the pMD19-T vector (TaKaRa, Dalian, China) at 16 °C and transformed into E. coli DH5α-competent cells (TIANGEN, Beijing, China). The 200 μL of transformation product was spread on LB solid medium plates containing ampicillin (Solarbio, Beijing, China) and incubated upside down at 37 °C for approximately 16 h. Eight independent positive clones were arbitrarily selected and sequenced by Xi’an Qingke Biological Company, Xi’an, China.

### 4.5. Bioinformatics Analysis of the Kiss1 Gene in HZ Boars

The open reading frame (ORF) of the *Kiss1* gene was searched by NCBI ORF finder (https://www.ncbi.nlm.nih.gov/orffinder/) (accessed on 15 May 2023), and the nucleotide sequence was aligned with the porcine *Kiss1* sequence downloaded on the NCBI database using the online BLAST algorithm (http://blast.ncbi.nlm.nih.gov/Blast.cgi) (accessed on 15 May 2023). The amino acid sequence homology between HZ pigs and other species was calculated by DNAMAN 8.0 software (Lynnon Biosoft, San Ramon, CA, USA). Basic physicochemical properties of the deduced Kiss1 protein were analyzed using the online site Protaram (https://web.expasy.org/protparam/) (accessed on 15 May 2023). The secondary and tertiary structures of the Kiss1 protein were obtained using the online tools SOPMA (https://npsa-prabi.ibcp.fr/cgi-bin/npsa_automat.pl?page=/NPSA/npsa_sopma.html) (accessed on 15 May 2023) and SWISS-MODEL (https://swissmodel.expasy.org) (accessed on 16 May 2023), respectively. Protein signal peptide was predicted by the online SignalP 4.1 tool (https://services.healthtech.dtu.dk/service.php?SignalP-4.1) (accessed on 16 May 2023). The transmembrane structure was predicted by the TMHMM 2.0 online program (https://services.healthtech.dtu.dk/service.php?TMHMM-2.0) (accessed on 16 May 2023). Potential proteins interacting with the HZ Kiss1 protein were searched through the STRING 11.0 database (https://string- db.org) (accessed on 16 May 2023). The conserved domain of Kiss1 protein was predicted by the CD-search online tool on the NCBI website (http://www.ncbi.nlm.nih.gov/Structure/cdd/wrpsb.cgi) (accessed on 16 May 2023).

### 4.6. Quantitative Real-Time PCR (qRT-PCR) Assay of the Kiss1 Gene in HZ and LC Boars

The *Kiss1* and *β-actin* (internal reference gene) expressions were detected using qRT-PCR on a LightCycler 480 II (Roche Applied Science, Mannheim, Germany). The PCR reaction system contained:10 μL 2 × SYBR^®^ Green Pro Taq HS Premix II (Accurate Biology, Changsha, China), 0.4 μL sense primers, 0.4 μL antisense primers, 2 μL cDNA template, and 6.4 μL RNase free H_2_O. The relative expression level of *Kiss1* mRNA was calculated through the 2^−∆∆Ct^ approach.

### 4.7. Western Blot Analysis of the Kiss1 Protein in HZ and LC Boars

Total protein from each testicular tissue was extracted with a radio immunoprecipitation assay (RIPA) protein extraction kit (Solarbio, Beijing, China), and the concentration of extracted protein was determined using a commercial bicinchoninic acid (BCA) assay kit (Solarbio, Beijing, China). The denatured proteins (20 µg) were separated by 10% sodium dodecyl sulfate polyacrylamide gel electrophoresis (SDS-PAGE) and subsequently were transferred to the polyvinylidene difluoride (PVDF) membranes (Solarbio, Beijing, China). The PVDF membranes were blocked with phosphate buffered saline tween-20 (PBST) containing 5% skim milk and then incubated with rabbit anti-Kiss1 polyclonal antibody (1:1000; Affinity, Nanjing, China) at 4 °C overnight. After washing, the membranes were incubated with goat anti-rabbit t IgG/HRP secondary antibodies (1:5000; Servicebio, Wuhan, China) for 2 h at 37 °C. The positive signals of the target proteins were visualized using an enhanced chemiluminescence (ECL) kit (Servicebio, Wuhan, China), and analyzed gray levels using ImageJ2 software (National Institutes of Health, New York, NY, USA). The β-actin band intensity was used to normalize.

### 4.8. Immunohistochemistry of the Kiss1 Protein in HZ and LC Boars

Paraffin sections were washed three times with distilled water after dewaxing and dehydration. Antigen sites were repaired and then endogenous peroxidase activity was eliminated with 3% H_2_O_2_. Sections were incubated with rabbit polyclonal anti-Kiss1 antibody (1:100; Affinity, Nanjing, China) overnight at 4 °C and phosphate buffered saline (PBS) instead of the primary antibody as the negative control. The positive signals (brown) were visualized with a diaminobenzidine (DAB) kit (Servicebio, Wuhan, China), and then observed using a Sunny EX31 biological microscope (Sunny, Ningbo, China).

### 4.9. Data Statistics

All the data were statistically analyzed using one-way analysis of variance (ANOVA) analysis in SPSS 26.0 software (SPSS, Chicago, IL, USA). All results are expressed as mean ± standard deviation (SD), with *p* < 0.05 and *p <* 0.01 considered as significant and extremely significant differences, respectively.

## 5. Conclusions

In summary, this is a study regarding the molecular cloning and characterization of the *Kiss1* CDS region in HZ boars and explores and compares its expression patterns and immunolocalization features during testicular development in HZ and LC boars. Histomorphology analysis revealed that boar testes gradually increased with age, and subsequently remained largely constant and that HZ boars develop faster than LC boars. The mRNA and encoded protein of the *Kiss1* gene are expressed during postnatal testis development in both HZ and LC boars, and their expression patterns are almost identical with higher expression levels in post-puberty than in pre-puberty. Kiss1 protein is predominantly present in the epithelia of seminiferous tubules and Leydig cells of pre-puberty, as well as in elongated spermatids, round spermatids, and spermatozoa of post-puberty. These results indicated that postnatal testicular development was faster in HZ boars than in LC boars and that Kiss1 may play an important regulatory role in the onset of puberty and spermatogenesis in boars, as well as providing additional insights into the differences in sexual function between HZ and LC boars.

## Figures and Tables

**Figure 1 ijms-24-16700-f001:**
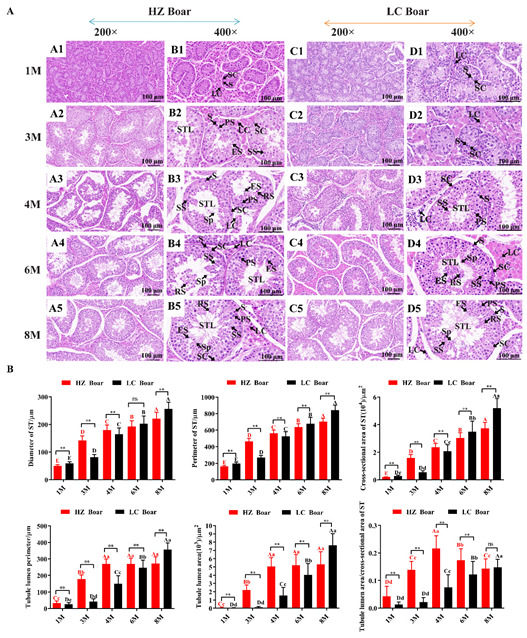
Morphological comparisons of testicular tissues in the HZ and LC boars at different development stages. (**A**) The representative photographs from 1 M, 3 M, 4 M, 6 M, and 8 M testicular cross-sections of HZ and LC boars at 200× and 400× magnification, respectively. (**B**) Morphological parameter values derived from testicular tissue of different age groups. (**A1**–**A5**) are representative photographs of 1 M, 3 M, 4 M, 6 M, and 8 M testis cross-sections of HZ boars at 200× magnification, respectively; (**B1**–**B5**) are representative photographs of 1 M, 3 M, 4 M, 6 M, and 8 M testis cross-sections of HZ boars at 400× magnification, respectively; (**C1**–**C5**) are representative photographs of 1 M, 3 M, 4 M, 6 M, and 8 M testis cross-sections of LC boars at 200× magnification, respectively; (**D1**–**D5**) are representative photographs of 1 M, 3 M, 4 M, 6 M, and 8 M testis cross-sections of LC boars at 400× magnification, respectively. S—spermatogonia; LC—Leydig cell; SC—Sertoli cell; STL—seminiferous tubule lumen; PS—primary spermatocyte; SS—secondary spermatocyte; RS—round spermatid; ES—elongated spermatid; Sp—spermatozoa. Data were presented as mean ± SD in the graphs. Different capital and lower letters represent extremely significant differences (*p* < 0.01) and significant differences (*p* < 0.05) between groups, respectively. Red letters denote comparisons between HZ boar groups, while black letters denote comparisons between LC boar groups. **: *p* < 0.01, and ns (no significance): *p* > 0.05. 1 M—1 month old; 3 M—3 months old; 4 M—4 months old; 6 M—6 months old; 8 M—8 months old.

**Figure 2 ijms-24-16700-f002:**
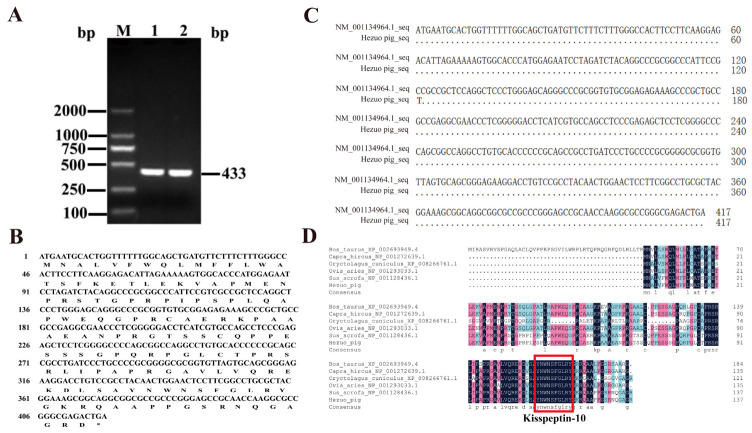
Cloning and sequence analysis of the coding sequence (CDS) of the HZ boars *Kiss1* gene. (**A**) PCR amplification products of the HZ boars *Kiss1* CDS sequence. M, DL 2000 marker; 1 and 2, *Kiss1* RT-PCR product. (**B**) Comparison of Kiss1 nucleotide sequence in HZ boar and reference sequence. (**C**) The nucleotide and deduced amino acid sequences for the cloned *Kiss1* CDS region. (**D**) Alignment of the deduced amino acid sequences of HZ boars *Kiss1* with that of cattle (accession no: XP_002693949.4), goat (accession No. NP_001272639.1), rabbit (accession No. XP_008266761.1), sheep (accession No. NP_001293033.1), and pig (accession No. NP_001128436.1).

**Figure 3 ijms-24-16700-f003:**
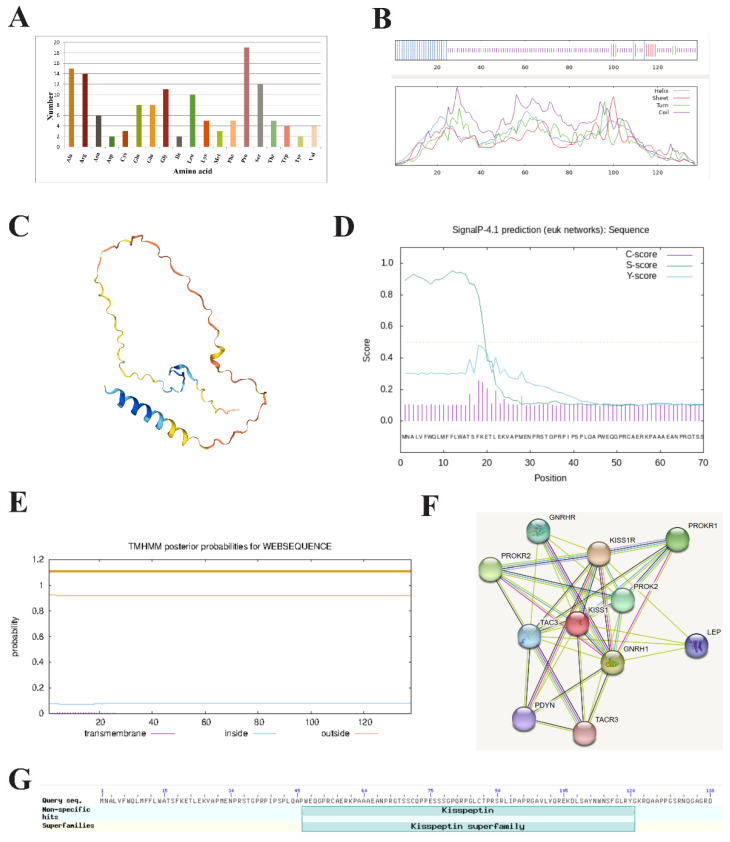
Bioinformatics analysis of HZ boar Kiss1 protein. (**A**) Analysis of the amino acid composition of HZ boars Kiss1 protein. (**B**) The secondary (**C**) and tertiary structure prediction of the HZ boars Kiss1 protein. (**D**) Signal peptide (**E**) and transmembrane structure prediction of the HZ boars Kiss1 protein. (**F**) Analysis of protein networks interacting with HZ boars Kiss1 protein. Network nodes represent the proteins encoded by the genes and edges represent protein–protein associations. (**G**) Conserved domain prediction of the HZ boars Kiss1 protein.

**Figure 4 ijms-24-16700-f004:**
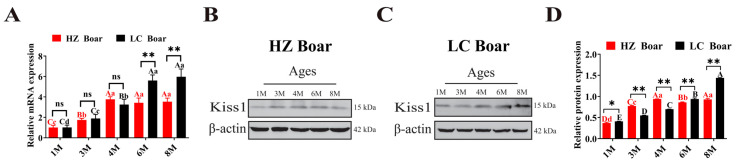
Temporal expression patterns of Kiss1 in HZ and LC boar testes. (**A**) Relative *Kiss1* mRNA expression. (**B**,**C**) Represent Western blot results of HZ and LC pigs, respectively. (**D**) Relative Kiss1 protein expression. β-actin was used as an internal reference gene. The results indicate the means ± SD. Different capital and lower letters represent extremely significant differences (*p* < 0.01) and significant differences (*p* < 0.05) between groups, respectively. Red letters denote comparisons between HZ boar groups, while black letters denote comparisons between LC boar groups. **: *p* < 0.01, *: *p* < 0.05, and ns (no significance): *p* > 0.05. 1 M—1 month old; 3 M—3 months old; 4 M—4 months old; 6 M—6 months old; 8 M—8 months old.

**Figure 5 ijms-24-16700-f005:**
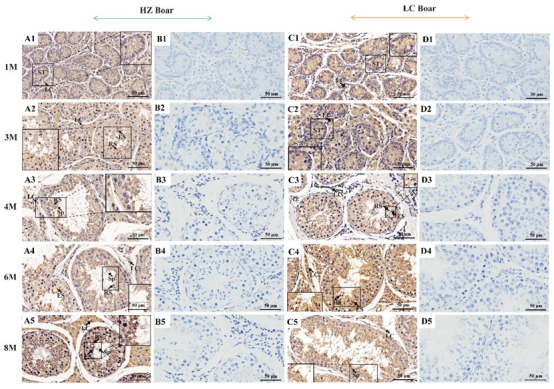
Immunohistochemical staining of Kiss1 protein in testes of HZ and LC boars at different developmental stages. (**A1**–**A5**) Immunostaining patterns of Kiss1 protein in the 1 M, 3 M, 4 M, 6 M, and 8 M HZ boar testes, respectively (400×); (**B1**–**B5**) Replace primary antibody with PBS as HZ boar testes negative control (400×); (**C1**–**C5**) Immunostaining patterns of Kiss1 protein in the 1 M, 3 M, 4 M, 6 M, and 8 M LC boar testes, respectively (400×); (**D1**–**D5**) Replace primary antibody with PBS as LC boar testes negative control (400×). LC—Leydig cell; RS—round spermatid; ES—elongated spermatid; Sp—spermatozoa, ST—seminiferous tubule. 1 M—1 month old; 3 M—3 months old; 4 M—4 months old; 6 M—6 months old; 8 M—8 months old.

**Table 1 ijms-24-16700-t001:** Information of primer sequence.

Gene	Sequence (5′-3′)	Length/bp	Utilization	Accession No.
*Kiss1*	F: CCAGGATGAATGCACTGGTT	433	cloning	NM_001134964.1
	R: GTTTGAAGGTCTCAGTCTCG			
*Kiss1*	F: ATGCACTGGTTTTTTGGCAGCTGAT	98	qRT-PCR	NM_001134964.1
	R: TGTAGATCTAGGATTCTCCATGGGT			
*β-actin*	F: ATATTGCTGCGCTCGTGGT	148	qRT-PCR	XM_003124280.5
	R: TAGGAGTCCTTCTGGCCCAT			

## Data Availability

Data is contained within the article.

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
