# Peer review of "Regulatory Effects of the Kiss1 Gene in the Testis on Puberty and Reproduction in Hezuo and Landrance Boars"

_ijms, 2023, doi:10.3390/ijms242316700_

Round 1

Reviewer 1 Report

Comments and Suggestions for Authors

The present manuscript compared the testicular expression of Kiss gene and protein in Hezuo (local Chinese breed) and Landrace boars during puberty. The histological study was oriented to demonstrate the modifications induced by puberty.

The topic is of interest but some issues that reduce the scientific relevance of the manuscript should be, in my opinion, addressed.

The title poorly reflects the complexity of the manuscript content. Some aspects are not immediately clear (i.e. puberty period, etc.) for the reader, thus the content of the manuscript appears, on the basis of the title, not very attractive.

Usually, kisspeptin is studied together with the expression of its receptor (kiss1 receptor). This is to demonstrate the local biological activity of the hormone. This aspect is completely omitted.

The puberty was only deduced on the basis of testicular histology. The modifications due to puberty cover a wide period, thus it is not possible to confirm the onset of puberty based only on this parameter. Unfortunately, other criteria were not considered (i.e. concentration of testosterone, presence/morphology/activity of epididymal spermatozoa).

Puberty and its modifications (anatomy, endocrinology) in poorly considered in the introduction. I suggest implementing this section.

In Figure 4, the increase in density of the band is evident in Landrace boar. In Hezou is not so clear, thus I suggest choosing a different image.

The number of animals included in the study (considering the number of groups) is limited. Furthermore, there is no evidence about how many portions of the testis were considered to perform the analysis. Did the authors select a specific point, similar in all the testicles? Or did the authors perform a pool of different portions? Furthermore, how many slides were prepared from each testis in for histology or immunohistochemistry? Did the authors perform a comparison between left and right testis? In other words, the authors should provide evidence that the analysis is representative of the whole testis.

References should be accurately revised. Not all studies on this matter are considered. Furthermore, some mistakes are present in the reference list (i.e. ref 26 and 27 is the same manuscript).

Reviewer 2 Report

Comments and Suggestions for Authors

In the article, Haixia et al. analyzed and compared morphological features and the expression and localization of Kiss1 during testicular development in Hezou and Landrace boars.

The topic of this work is original and relevant to the field, as it uncovers the first evidence that Kiss1 can regulate the onset of puberty and spermatogenesis in Hezou boars and provides greater insight into differences in sexual function between Hezou and Landrace boars.

By using a number of research methods, such as H&E staining, cloning, immunohistochemistry, qRT-PCR and Western blot, the authors showed that Hezou boars mature faster than Landrace boars, and the postnatal testicular development of Hezou boars is also faster. At the same time, they showed that Kiss1 may play an important regulatory role in the initiation of puberty and spermatogenesis in boars, especially during spermiogenesis.

The authors obtained the results, thanks to which they could open a very interesting discussion, bringing a lot of new and interesting information closely related to the issues raised by them. For example, boar Kiss1 gene has a base mutation in the sequence of the CDS region, and whether it is associated with precocious puberty in Hezou pigs requires further study.

In my opinion, the materials and research methods mentioned and used are adequate and do not raise any major objections. The conclusions are consistent with the evidence, and the arguments presented respond well to the main question posed. The thoroughly researched and up-to-date literature, with which the authors skillfully discuss, deserves praise. The value of the work increases especially thanks to the very well presented and described figures.

Shouldn't the content contained in lines 68-70 be included in the discussion or conclusions?
